# Influence of DNA Mismatch Repair (MMR) System in Survival and Response to Immune Checkpoint Inhibitors (ICIs) in Non-Small Cell Lung Cancer (NSCLC): Retrospective Analysis

**DOI:** 10.3390/biomedicines10020360

**Published:** 2022-02-02

**Authors:** Alejandro Olivares-Hernández, Edel del Barco Morillo, Carmen Parra Pérez, José Pablo Miramontes-González, Luis Figuero-Pérez, Teresa Martín-Gómez, Roberto Escala-Cornejo, Lorena Bellido Hernández, Rogelio González Sarmiento, Juan Jesús Cruz-Hernández, María Dolores Ludeña de la Cruz

**Affiliations:** 1Department of Medical Oncology, University Hospital of Salamanca, 37007 Salamanca, Spain; ebarco@saludcastillayleon.es (E.d.B.M.); lfiguero@saludcastillayleon.es (L.F.-P.); mtmartingo@saludcastillayleon.es (T.M.-G.); lbellido@saludcastillayleon.es (L.B.H.); jjcruz@usal.es (J.J.C.-H.); 2Institute for Biomedical Research of Salamanca (IBSAL), 37007 Salamanca, Spain; gonzalez@usal.es; 3Department of Medicine, University of Salamanca, 37007 Salamanca, Spain; mdludena@saludcastillayleon.es; 4Department of Pathology, University Hospital of Salamanca, 37007 Salamanca, Spain; 5Department of Internal Medicine, University Hospital Rio Hortega, 47012 Valladolid, Spain; jpmiramontes@hotmail.com; 6Department of Medicine, University of Valladolid, 45005 Valladolid, Spain; 7Instituto de Investigación Nacional (SOLCA) de Guayaquil, Guayaquil 090514, Ecuador; roberto.a.escala@solca.med.ec

**Keywords:** MMR system, immunotherapy, ICIs, NSCLC, dMMR/MSI-H, survival, response

## Abstract

Mutations in the mismatch repair (MMR) system predict the response to immune checkpoint inhibitors (ICIs) like colon or gastric cancer. However, the MMR system’s involvement in non-small cell lung cancer (NSCLC) remains unknown. Addressing this issue will improve clinical guidelines in the case of mutations in the main genes of the MMR system (MLH1, MSH2, MSH6, and PMS2). This work retrospectively assessed the role that these gene mutations play in the response to and survival of ICIs in NSCLC. Patients with NSCLC treated with nivolumab as the second-line treatment in the University Hospital of Salamanca were enrolled in this study. Survival and response analyses were performed according to groups of MMR system gene expression (MMR expression present or deficiency) and other subgroups, such as toxicity. There was a statistically significant relationship between the best response obtained and the expression of the MMR system (*p* = 0.045). The presence of toxicity grade ≥ 3 was associated with the deficiency expression of MMR (dMMR/MSI-H) group (*p* = 0.022; odds ratio = 10.167, 95% confidence interval (CI) 1.669–61.919). A trend towards greater survival and response to ICIs was observed in NSCLC and dMMR. Assessing the genes in the MMR system involved in NSCLC is key to obtaining personalized immunotherapy treatments.

## 1. Introduction

Immunotherapy has now revolutionized the treatment of solid tumors, especially since the introduction of ipilimumab, the first drug approved by the Food and Drug Administration (FDA) in 2011 for the treatment of advanced unresectable melanoma [1]. Immune checkpoint inhibitors (ICIs) have increased survival rates for non-small cell lung cancer (NSCLC) by up to 20–30% to five years, where previously the overall survival rates (OS) did not reach beyond 12–18 months [2]. However, one of the current challenges in oncology is the search for patients with long responses to immunotherapy, which would allow for precision medicine and higher numbers of long-term survivors. The discovery of different predictive biomarkers that respond to ICIs will revolutionize precision medicine. Different genomic predictive biomarkers, such as the KRAS, TP53, or STK11 genes, have been studied in this field with contradictory results [3,4,5]. Currently, one of the most promising avenues for the treatment of solid tumors and the most unknown element of NSCLC is the mismatch repair (MMR) system genes.

Around 4–5% of NSCLCs show alterations in the genes that make up the MMR system (deficiency of expression, dMMR/MSI-H) [6]. The MMR system is one of the most important guardians of genomic integrity. It improves the fidelity of DNA replication, aborts illegitimate recombination, and affects the outcome of several other processes of DNA metabolism. The malfunction of MMR gives rise to a mutator phenotype and microsatellite instability [7]. The role of MMR system defects in the response to immunotherapy is well known and has been widely studied in different tumors, mainly at the gastrointestinal (GI) tract level [8]. Different studies based on patient series or retrospective analysis have shown that the deficiency expression of MSH2 or MLH1 in lung adenocarcinomas has been associated with poor survival of these patients due to resistance to immunotherapy [9]. The deficiency of this repair system has been observed to be more significant in tumors of the adenocarcinoma or squamous cell lineages compared to a microcytic histology. In most cases, the silencing of the genes of the MMR system is due to epigenetic alterations and not so much to mutations [10].

Mutations of MMR genes have been associated with mutations of the KRAS or EGFR genes in NSCLC patients; however, this association has not translated into a prognostic or predictive implication on the response to chemotherapy or immunotherapy [11]. No studies have assessed the relationship between NSCLC and dMMR/MSI-H tumors’ response to immunotherapy. There are only data based on single case reports or case series and other data based on extrapolations from larger studies that included more tumors, such as studies conducted by Zhao et al. [6] and Viale et al. [9].

Therefore, although the MMR system and its implications for immunotherapy for NSCLC have not been clearly studied, this work assumes that, as in other tumors, such as colon cancer, its presence is a favorable predictive factor for response to these treatments. Therefore, immunotherapy may be considered in a previous line to the foreseen or in patients with conductive mutations [12]. Future studies can determine how the presence of these alterations in the MMR genes can affect immunotherapy for NSCLC. As such, the objective of this work is to evaluate mutations in the MMR system and their effects on the response to immunotherapy for NSCLC and subsequent survival.

## 2. Material and Methods

### 2.1. Data Collection and Construction of the Cohort

A retrospective hospital-based study was carried out on selected patients treated at the Department of Medical Oncology in the Complejo Asistencial Universitario de Salamanca (Salamanca, Spain). The inclusion criteria were as follows:Patients diagnosed with advanced or metastatic NSCLC.The patients must have received treatment with ICIs (Nivolumab) between 2015 and 2021.Second-line immunotherapy treatment with an anti-PD1 drug (nivolumab independent of PD-L1 expression in the tumor, according to clinical trials CheckMate 017 and CheckMate 057) [13,14].The existence of adequate samples from each patient in the Pathology Department of de University Hospital of Salamanca for IHC analysis.Patients over 18 years with full mental faculties who received and signed the informed consent for inclusion in the research project. After the age of 18, there is no age limit.Patients had to present an Eastern Cooperative Oncology Group (ECOG) 0-1.

The exclusion criteria were as follows:Refusal for inclusion in the research project by the patient or if age under 18 years old.Small cell lung cancer.First-line treatment with immunotherapy.No sample of primary tumor.ECOG ≥ 2

The fundamental data collected and studied were age (years), sex, histology, sites of metastases, progression-free survival (PFS) in months, overall survival (OS) in months, number of doses received, and the best response obtained with immunotherapy and immunotoxicity. The study was carried out according to the ethical protocols of the hospital and after obtaining informed consent from patients for the extraction of the samples. The patients signed informed consent that allowed the study of their primary tumor samples for different research purposes, as was done later this study. If the information found in this study is favourable for inclusion in immunotherapy treatment regimens, it will be included in the NSCLC treatment protocols of the Complejo Asistencial Universitario de Salamanca.

Subsequently, the previous data were analyzed together with the expression data of the MMR system for the MLH1, MSH2, MSH6, and PM2 genes. An analysis of survival and response was carried out in patients based on the mutations found in the four previous genes. The patients were subdivided by positive or negative PD-L1 expression (this subdivision was made because it is the most widely used in clinical trials for NSCLC and second-line immunotherapy, such as the CheckMate 017 and CheckMate 057 trials). To carry out this classification of patients, we relied on published clinical trials of second-line immunotherapy, in which the main factor for dividing the subgroups was the positive or negative expression of PD-L1.

### 2.2. Analysis of MMR System Expression by IHC

The expression analysis of the MMR system was performed with the peroxidase anti-peroxidase immunohistochemistry (IHC) technique using Leica BOND Polymer development kits (Buffalo Grove, IL 60089 United States). Leica BOND III automatic machines (Buffalo Grove, IL 60089 United States) were subsequently used. We carried out a semi-quantitative study via analysis at 10 fields of 20×. The samples were evaluated by two independent doctors to avoid bias. Microscopic analysis was carried out with Nikon Eclipse Ci microscopy equipment. All of the samples analyzed by IHC were from NSCLC primary tumors, and were carried out during the year 2020 after collecting data from all patients.

### 2.3. Statistical Analysis

For the statistical analysis, PFS was first calculated as the months from the initiation of immunotherapy treatment to the clinical or radiological progression. OS was calculated as the period (in months) from the initiation of the patient’s immunotherapy treatment until death. Survival rates were calculated as medians with 95% confidence intervals (CIs). Survival and response as functions of MMR expression were calculated using the Kaplan–Meier method (log rank and Breslow test), Cox regression analysis, and chi-squared distribution. Subgroup analyses were performed for sex, histology, and the expression of PD-L1 to avoid confounding factors. All survival data are expressed as medians. The statistical significance for the analyses in this study was established at *p* < 0.05. The software used was SPSS version 25 (IBM). 

## 3. Results

### 3.1. General Analysis of the Sample

A total of 73 NSCLC samples were studied (Table 1). The median age of the patients was 68 years (44–84). There were 59 males (80.8%) and 14 females (19.2%). The most frequent histology was adenocarcinoma in 36 patients (49.3%), followed by squamous cell cancer in 34 patients (46.6%). The most frequent sites of metastatic involvement were the lung (45 patients, 61.6%) and lymph nodes (39 patients, 53.4%), followed by the bone (14 patients, 19.2%) and liver (10 patients, 13.7%). The PD-L1 expression was negative (0%) in 27 patients and positive (≥1%) in 46 patients. Loss of expression of the repair genes was observed in 6 of the 73 patients analyzed (8.2%). In the six patients in whom the loss of expression was observed, the affected gene was PMS2. In addition, for one patient, a double loss of expression was observed as a deficit in MLH1 and PMS2. In the total sample, no driver mutations were found.

The OS of the sample was 13 months (95% CI 8.2–17.8). The PFS was 5 months (95% CI 3.8–6.2). The response results were as follows: (1) 4 patients (5.4%) had a complete response, (2) 13 patients (17.8%) had a partial response, (3) stabilization was observed in 16 patients (21.9%), and (4) progression was observed in 40 patients (54.8%). Seven was the mean number of doses administered. The observed toxicity was grade ≥ 3 in 8 patients (CTCAE v5.0).

### 3.2. Survival Analysis by Expression of the MMR System

The OS of patients with a preserved expression was 12 months versus 14 months for those with tumors who had a loss of expression of the MMR system (*p* = 0.598; Figure 1). The PFS of patients with a preserved expression of the MMR system was 4 months versus 8 months for patients with a loss of expression (*p* = 0.661; Figure 2).

No differences in OS were observed based on the expression of the MMR system based on factors such as sex (only males presented loss of expression, *p* = 0.595), histology (adenocarcinoma group, *p* = 0.359, and epidermoid group, *p* = 0.271) or PD-L1 positivity/negativity (PD-L1 negative group, *p* = 0.744, and positive, *p* = 0.641). No differences in PFS were observed in MMR conservation or expression groups based on factors such as sex (*p* = 0.668), histology (adenocarcinoma, *p* = 0.399, and epidermoid, *p* = 0.556), or PD-L1 (negative group, *p* = 0.753, and positive, *p* = 0.346). 

### 3.3. Analysis of Response by Expression of the MMR System

The existence of a relationship between the loss of expression of the MMR system and the best response obtained (four groups based on RECIST 1.1 criteria) with immunotherapy treatment using a Pearson chi-squared test was analyzed. The results showed a significance of *p* = 0.045. Figure 3 shows the relationship between the objective response rate (ORR) and the expression of the MMR system in absolute numbers of the total number of patients in each subgroup.

We analyzed whether the presence of severe toxicity (grade ≥ 3) was correlated with a loss of expression of repair genes. A comparison was made using Fisher’s exact test, where *p* = 0.022 was observed (odds ratio (OR) = 10.167 (95% CI 1.669–61.919)). Therefore, there was a statistically significant association between toxicity ≥ 3 and loss of expression of MMR system genes. The calculation was then adjusted according to the best response obtained as a confounding factor, dividing the response groups according to responses of yes or no. The association between toxicity and the IHC of the MMR system was focused on those patients with favorable responses (stabilization or partial or complete response).

## 4. Discussion

The expression of the repair system has seldom been studied until now, and therefore, its implications for the treatment of cancers and the prevalence of these alterations are not clearly known [15]. Different studies have calculated the prevalence of mutations in the genes of the MMR system in lung cancer to be 4–5%. However, it is estimated that the reduction in the expression of these genes could be even greater at upwards of 50% in different series (Xinarianos et al. and Hsu et al.) [16,17]. In our series, the loss of expression of the repair genes was 8.2%, which is higher than that shown in other studies. To date, studies have shown that the most altered gene in these cases is MSH2. This is contrary to our study, which did not show that there were alterations in this gene, with the most frequent alteration being in PMS2 [18]. 

The survival rates assessed were different in the cases of OS and PFS. In the case of OS, no differences were found between the two groups evaluated. On the contrary, PFS was doubled in patients with a loss of expression of the repair genes (8 months) compared to those with preserved expression (4 months). Although the results are not statistically significant, there is a clear trend towards higher PFS in patients with a loss of expression. This is in accordance with different studies carried out on other solid tumors. The KEYNOTE-177 study, conducted in patients with metastatic colon cancer and dMMR/MSI-H, showed a PFS of 16.5 months in patients treated with immunotherapy versus 8.2 months in those treated with chemotherapy and capecitabine-based regimens [19]. The current study marks the beginning of immunotherapy research on solid tumors with alterations in the MMR system, and it is possible that in patients with NSCLC, this benefit will also be demonstrated in the future. This is why, despite the main limitation of the study, which is the low prevalence of mutations in the MMR system in NSCLC, the results showed a clear tendency towards a better response to immunotherapy in these patients. The implications of the MMR system genes in NSCLC could be assessed in patients with a PD-L1 expression < 50% in whom current treatment is based on a combination of platinum-based chemotherapy plus immunotherapy [20,21]. In the future, monotherapy immunotherapy could be applied in dMMR/MSI-H tumors, avoiding the toxicity associated with unnecessary chemotherapy [22]. 

Due to the limitation of the sample size, the association between the loss of expression and improved response to immunotherapy was evaluated. When the association between the two qualitative variables was evaluated, there was a statistically significant relationship based on the results of a chi-squared test. This relationship shows how cases with a loss of expression of the repair genes present a greater response, primarily at the expense of patients with partial responses [23]. These observations were not influenced by other factors such as age, sex, or histological subtype. This is why, even with the low prevalence of the loss of expression of reparative genes in NSCLC, the performance of these routine genes in patients with metastatic or advanced NSCLC should be evaluated through a cost-efficiency study. IHC could realize such a low-cost study, and it would be possible for many patients to benefit thanks to a prevalence of around 5%, which, according to some series, could be even higher [24].

In addition, whether the presence of a greater toxicity was related to a loss of expression of the repair genes was also analyzed. It was observed that a higher toxicity (divided into two subgroups depending on whether to present a toxicity grade ≥ 3 or severe) was related to a loss of expression of the repair genes. However, despite this association, it was verified that if there was a confounding factor, such as the response obtained, the response was the one associated with toxicity, and this, in turn, led to the loss of expression of the repair genes. Therefore, this opens a new avenue of study, as the association between the response to immunotherapy and toxicity should be clarified [25]. To date, multiple studies have assessed the more than likely existence of a causal association between increased response and severe toxicity ≥2 or 3 [26]. Thus far, no study has clearly defined this strength of association; however, it is likely that the increased activity of the immune system against tumors leads to a greater antitumor response, followed by the consequent immunorelational toxicity secondary to this immune activation at the expense of T lymphocytes [27]. 

To conclude, it is important to assess whether NSCLC should be included in the sphere of Lynch syndrome [28]. Tobacco’s clear association with NSCLC may have taken a backseat to the genetic causes of lung cancer [29]. However, in the future, 10% of patients with non-smoking NSCLC should be included in different molecular studies, which will show that NSCLC’s origin is more significant than the already known oncogenic drivers of lung cancer, such as EGFR, ALK, or ROS1 [30].

## 5. Conclusions

Alterations of MMR system genes in NSCLC appear to be associated with an increased response to immunotherapy, similar to other solid tumors. In the case of these tumors, mutations in the MMR system and losses of expression must be studied in the future. New studies and clinical trials that assess these conclusions will be key for precision medicine for patients with NSCLC treated with immunotherapy. 

## Figures and Tables

**Figure 1 biomedicines-10-00360-f001:**
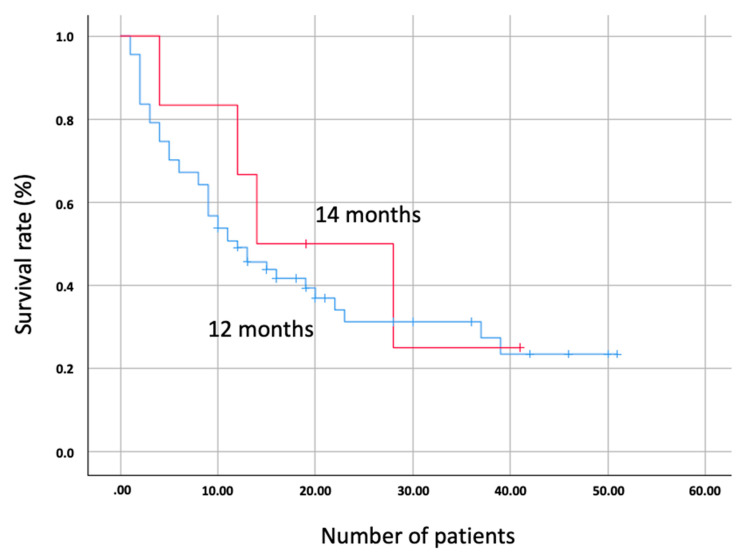
Kaplan–Meier curve showing the comparison of overall survival (OS) between patients with lost mismatch repair (MMR) system gene expression (red) and conserved (blue).

**Figure 2 biomedicines-10-00360-f002:**
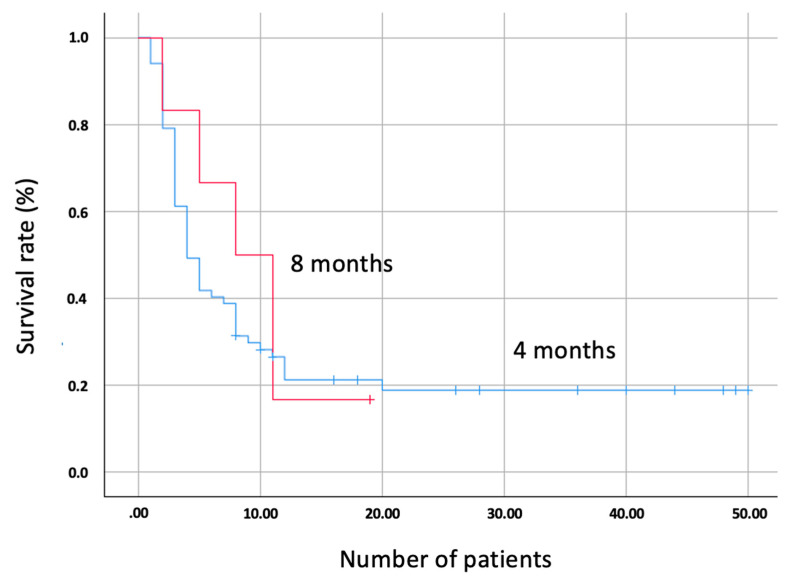
Kaplan–Meier curve showing the comparison of progression-free survival (PFS) between patients with lost MMR system gene expression (red) and conserved (blue).

**Figure 3 biomedicines-10-00360-f003:**
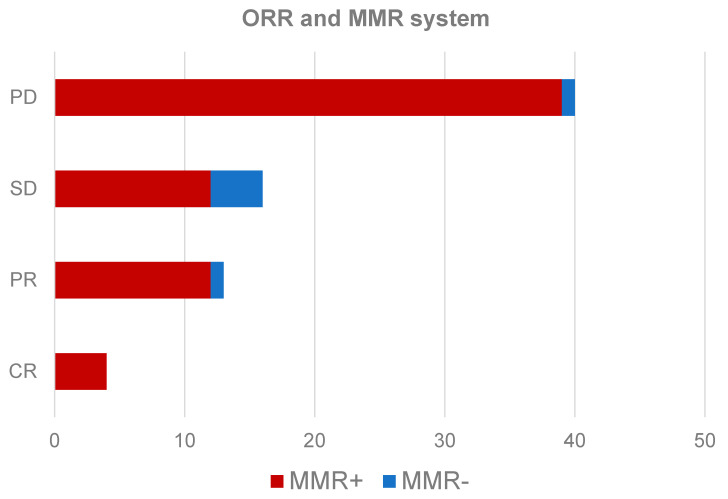
Relationship between objective response rate (ORR) and mutations in the MMR system in patients treated with immunotherapy in NSCLC. MMR+, present expression of MMR system; MMR-, deficiency expression of MMR system; PD, progressive disease; SD, stable disease; PR, partial response; CR, complete response.

**Table 1 biomedicines-10-00360-t001:** General characteristics of the sample. The table shows the demographic variables of the study population of patients with advanced or metastatic NSCLC under treatment with immunotherapy at the Complejo Asistencial Universitario de Salamanca.

Sample	Overall, 73 (100%)	MMR Present, 67 (91.8%)	MMR Deficiency, 6 (8.2%)
Age	68 (44–84)	68 (44–84)	67 (54–79)
Sex (M/W)	59/14 (80.8/19.2%)	53/14 (79.1/20.9%)	6/0 (100/0%)
Subtype			
ADC Squamous Undifferentiated	36 (49.3%)	32 (47.8%)	4 (66.7%)
	34 (46.6%)	32 (47.8%)	2 (33.3%)
	3 (4.1%)	3 (4.5%)	0 (0%)
PD-L1			
Negative Positive	27 (37%)	25 (37.3%)	3 (50%)
	46 (63%)	42 (62.7%)	3 (50%)
Driver mutations			
EGFR, ALK, or ROS1	0 (0%)	0 (0%)	0 (0%)
Survival (months)			
Overall Progression free	13 (95% CI 8.2–17.8)	12 (95% CI 4–20)	14 (95% CI 4.5−23.5)
	5 (95% CI 3.8–6.2)	4 (95% CI 3–5)	8 (95% CI 2–14)
Response			
Progression or death Stabilisation Partial response Complete response	40 (54.8%)	39 (58.2%)	1 (16.7%)
	16 (21.9%)	12 (17.9%)	4 (66.7%)
	13 (17.8%)	12 (17.9%)	1 (16.7%)
	4 (5.4%)	4 (6%)	0 (0%)
Toxicity			
Not observed Asthenia Endocrine Dermal Gastrointestinal Hepatic Renal Cardiac Pulmonary	47 (64.4%)	45 (67.2%)	2 (33.3%)
	12 (16.4%)	11 (16.4%)	1 (16.7%)
	4 (5.8%)	2 (3%)	2 (33.3%)
	2 (2.7%)	1 (1.5%)	1 (16.7%)
	4 (5.8%)	3 (4.5%)	1 (16.7%)
	4 (5.8%)	2 (3%)	2 (33.3%)
	7 (9.6%)	5 (7.5%)	2 (33.3%)
	1 (1.4%)	1 (1.5%)	0 (0%)
	1 (1.4%)	1 (1.5%)	0 (0%)

## Data Availability

The data presented in this study are available on request from the corresponding author. The data are not publicly available due to are the property of the Medical Oncology Department of the University Hospital of Salamanca.

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
