# Peer review of "Influence of DNA Mismatch Repair (MMR) System in Survival and Response to Immune Checkpoint Inhibitors (ICIs) in Non-Small Cell Lung Cancer (NSCLC): Retrospective Analysis"

_biomedicines, 2022, doi:10.3390/biomedicines10020360_

Round 1
Reviewer 1 Report
This manuscript is a beneficial examination for exact medical developments of the immune checkpoint inhibitor of future lung cancer. We want to examine it to see results of figure 3 at another time.
It is useful and I think that it was able to be a valuable report by major revision. Here is a summary of steps that I would recommend:
Major revision:
1. At first, we can not see "Figure 3" at all. I am interested in contents of figure 3 very much.
2. In table 1, correct age and range of "Adenocaricnoma" and "Squamous" ,please. But what I want to expect in detail, the columns may be "Overall", "conserved MMR" and "lost MMR". May that table explain figure 1 and figure 2 easily?
3. How about are driver mutations status of overall patients, such as EGFR? Probably, I think that all patients are received chemotherapy as first line therapy because of without driver mutation. You need describe driver mutation status for interpretation of overall survival.
Minor revision:
1. P1 Abstract L10,12 What do you mean with dMMR? Probably, it means "defective mismatch repair (dMMR)". When you define dMMR with a text, I am glad.
2. P3 L22 Please revise "PDL-1" for "PD-L1".
3. P7 L20 Please revise "PDL-1" for "PD-L1".
Reviewer 2 Report
- Originality: Does the paper contain new and significant information adequate to justify publication?
The authors wrote a promising retrospective study in which they designed a projectile view of second-line treatment with nivolumab in dMMR NSCLC. In this way, the authors searched the current literature and tried to find out which paths are emerging in this field and where they are leading. On the other hand, they analysed the results in detail retrospectively in their clinic. There are also important findings by the authors that are discussed in the relevant part of the paper. As a result, the paper can be published with some revisions in which the authors clearly state and defend the search terms, survival tables and contradictions in terms.
- Relationship to Literature: Does the paper demonstrate an adequate understanding of the relevant literature in the field and cite an appropriate range of literature sources? Is any significant work ignored?
Since the paper is about deficiency of mismatch repair and efficacy of second line nivolumab in advanced stage non small cell lung cancer treatment, a smart and clear literature review is done with current studies. Also the topic is very emerging. The predictive role of dMMR / MSI-h status in lung cancer is not as well studied as the effect of PD-L1.
- Methodology: Is the paper's argument built on an appropriate base of theory, concepts or other ideas? Has the research or equivalent intellectual work on which the paper is based been well designed? Are the methods employed appropriate?
The methodology of the paper is quite common since the authors try to conduct literature review and bibliometric review. But, authors not only state frequency tables or descriptive results as well. They also try to visualize the results and clearly identify the possible relations with kaplan-meier diagrams. These visualizations make the discussion very clear and argumentative.
But they should correct the definitions of the x and y curves in the survival charts. The Y curve should show ‘’the survival rate’’ and the X curve should show ‘’the number of patients’’.
Moreover, the patient criteria included in the study should be specified in more detail. such as age range, mutation status, performance status, previous treatment modalities, the years between which patients are included.
On the other hand, chosen keywords of the whole research is not well defined though authors tried to explain the purpose briefly. For instance, why the combinations of the dMMR / MSI-H and NSCLC type acronyms were not used or didn’t the authors find any irrelevant paper after their searches are not defined well.
Lastly, It will be instructive for readers to briefly describe the status of dMMR and MSI-h in the introductory section of the article.
- Results: Are results presented clearly and analysed appropriately? Do the conclusions adequately tie together the other elements of the paper?
As I stated in methodology section, the visualizations of the results and findings are clear. But they should correct the definitions of the x and y curves in the survival charts. The Y curve should show ‘’the survival rate’’ and the X curve should show ‘’the number of patients’’. Also, when giving the survival results in the article, it should be stated whether it is a ‘’median or mean’’.
- Implications for research, practice and/or society: Does the paper identify clearly any implications for research, practice and/or society? Does the paper bridge the gap between theory and practice? How can the research be used in practice (economic and commercial impact), in teaching, to influence public policy, in research (contributing to the body of knowledge)? What is the impact upon society (influencing public attitudes, affecting quality of life)? Are these implications consistent with the findings and conclusions of the paper?
The phases of the conducted study are designed well. also, nearly all the applications are explained clearly. The theoretical implications are defined well.
- Quality of Communication: Does the paper clearly express its case, measured against the technical language of the fields and the expected knowledge of the journal's readership? Has attention been paid to the clarity of expression and readability, such as sentence structure, jargon use, acronyms, etc.
The communication quality of the paper is clear. Authors express what they do well. But, in at the beginning of the methodology they just skipped the keywords and possible combinations of the keywords they did not included to the methodology. Those exclusions should be clearly defined.
Reviewer 3 Report
The article “Role of the DNA Mismatch Repair (MMR) System in Survival and Response to Immunotherapy in Non-Small Cell Lung Cancer (NSCLC)” presents some interesting results of a retrospective analysis. The main issue of the study is that less than 10% of the patients had loss of expression of the repair genes, which makes the statistical analysis somewhat problematic. Would there be any possibility to include more patients?
Some other comments:
- The title should specify what kind of article this is (retrospective analysis) and perhaps the main finding.
- In the Material and Method section, the authors state that “A retrospective hospital-based study was carried out on selected patients treated at the Complejo Asistencial Universitario de Salamanca (Salamanca, Spain)” How were the patients selected (other than the inclusion criteria)? Also, the Study Description is a bit confusing. For example, the authors state that they obtained informed consent, but they do not explain for what, since this was a retrospective study. The period of time in which patients were included should be specified. The authors should clarify at which time point and on which type of tissue MMR and PDL1 analysis was performed.
- There is no figure where the legend for figure 3 appears
Minor comments:
- Abstract: “Mutations in the mismatch repair (MMR) system predict the response to immune check-
point inhibitors (ICIs), but the MMR system’s involvement in non-small cell lung cancer (NSCLC)
remains unknown.” The authors should specify for which tumors MMR predicts. Also, the abstract should be structured with Material and Methods and Results section for better comprehension.
- At the end of the Introduction section, the authors forgot to delete some of the text that comes with the template (This section may be divided by subheadings. It should provide a concise and precise description of the experimental results, their interpretation, as well as the experimental conclusions
that can be drawn.)
- English issues (revision by a native English speaker required). Some examples:
Introduction: “Immune checkpoint inhibitors (ICIs) have increased survival rates for non-small cell lung
cancer (NSCLC) by up to 20–30% to five years, despite the overall survival rates (OS) not previously reaching beyond 12–18 months [2].”, “Different studies based on patient series or retrospective levels have shown”, “this association has not translated into a prognostic or predictive implication for the response to chemotherapy (...)”, “Therefore, its initiation may be considered in a previous line to the one foreseen or in patients with conductive mutations [11].”
Round 2
Reviewer 1 Report
Dear Alejandro Olivares Hernández
I appreciate your polite answer. Thank you for revising a table. Thus, it became easy to understand very much.
We could see figure 3, and the significance of the study became more interesting.
I hope this study will be a medical developments of the immune checkpoint inhibitor of future lung cancer.
Thank you.
Reviewer 3 Report
The authors have answered all comments.